# APOBEC SBS13 Mutational Signature—A Novel Predictor of Radioactive Iodine Refractory Papillary Thyroid Carcinoma

**DOI:** 10.3390/cancers14061584

**Published:** 2022-03-21

**Authors:** Sarah Siraj, Tariq Masoodi, Abdul K. Siraj, Saud Azam, Zeeshan Qadri, Sandeep K. Parvathareddy, Rong Bu, Khawar S. Siddiqui, Saif S. Al-Sobhi, Mohammed AlDawish, Khawla S. Al-Kuraya

**Affiliations:** 1Human Cancer Genomic Research, King Faisal Specialist Hospital and Research Centre, P.O. Box 3354, Riyadh 11211, Saudi Arabia; sarah-siraj@kfshrc.edu.sa (S.S.); tmasoodi@kfshrc.edu.sa (T.M.); asiraj@kfshrc.edu.sa (A.K.S.); saazam@kfshrc.edu.sa (S.A.); sqadri96@kfshrc.edu.sa (Z.Q.); psandeepkumar@kfshrc.edu.sa (S.K.P.); rbu@kfshrc.edu.sa (R.B.); 2Department of Pediatric Hematology-Oncology, King Faisal Specialist Hospital and Research Center, P.O. Box 3354, Riyadh 11211, Saudi Arabia; ksiddiqui@kfshrc.edu.sa; 3Department of Surgery, King Faisal Specialist Hospital and Research Center, P.O. Box 3354, Riyadh 11211, Saudi Arabia; sobhi@kfshrc.edu.sa; 4Department of Endocrinology and Diabetes, Prince Sultan Military Medical City, P.O. Box 261370, Riyadh 11342, Saudi Arabia; maldawish@psmmc.med.sa

**Keywords:** papillary thyroid cancer, radioactive iodine refractory, COSMIC mutational signatures, SBS13, APOBEC-related mutations

## Abstract

**Simple Summary:**

Around 15% of papillary thyroid carcinoma (PTC) patients are not cured using standard surgery followed by radioactive iodine (RAI) therapy, and instead develop refractory disease. The aim of this study was to help identify RAI-refractory PTC patients early and guide precision medicine by performing a clinical and genomic characterization of RAI-refractory and avid PTCs. RAI-refractory PTCs had a more aggressive clinical presentation, a higher number of mutations, harbored more *TERT* promoter (*TERT*p) mutations, and were enriched with APOBEC-related mutations. Notably, the APOBEC single-base substitution (SBS) mutational signature, SBS13, and *TERT*p mutations were revealed to be independent predictors of RAI refractoriness in PTC. Although SBS13 and *TERT*p mutations alone highly predicted RAI refractoriness, when combined, they formed a stronger predictor of RAI refractoriness in PTC. This study highlights the APOBEC SBS13 mutational signature as a novel independent predictor of RAI refractoriness in a distinct subgroup of PTC.

**Abstract:**

Standard surgery followed by radioactive iodine (^131^I, RAI) therapy are not curative for 5–20% of papillary thyroid carcinoma (PTC) patients with RAI refractory disease. Early predictors indicating therapeutic response to RAI therapy in PTC are yet to be elucidated. Whole-exome sequencing was performed (at median depth 198x) on 66 RAI-refractory and 92 RAI-avid PTCs with patient-matched germline. RAI-refractory tumors were significantly associated with distinct aggressive clinicopathological features, including positive surgical margins (*p* = 0.016) and the presence of lymph node metastases at primary diagnosis (*p* = 0.012); higher nonsilent tumor mutation burden (*p* = 0.011); *TERT* promoter (*TERT*p) mutation (*p* < 0.0001); and the enrichment of the APOBEC-related single-base substitution (SBS) COSMIC mutational signatures 2 (*p* = 0.030) and 13 (*p* < 0.001). Notably, SBS13 (odds ratio [OR] 30.4, 95% confidence intervals [CI] 1.43–647.22) and *TERT*p mutation (OR 41.3, 95% CI 4.35–391.60) were revealed to be independent predictors of RAI refractoriness in PTC (*p* = 0.029 and 0.001, respectively). Although SBS13 and *TERT*p mutations alone highly predicted RAI refractoriness, when combined, they significantly increased the likelihood of predicting RAI refractoriness in PTC. This study highlights the APOBEC SBS13 mutational signature as a novel independent predictor of RAI refractoriness in a distinct subgroup of PTC.

## 1. Introduction

Papillary thyroid carcinoma (PTC) is the most prevalent endocrine malignancy both worldwide [1,2] and in the Middle East [3,4]. Under standard treatment (surgical intervention followed by adjuvant radioactive iodine [^131^I; RAI] to destroy residual disease, prevent tumor growth and/or recurrence), PTC has excellent prognosis [5,6]. Nevertheless, 5–20% of patients become refractory to RAI therapy, seemingly no longer responding or showing resistance to RAI therapy [7,8,9]. RAI refractoriness in PTC has been associated with poorer clinical outcome, increased mortality [10,11] and limited treatment options that are typically accompanied by considerable toxicity [12,13]. The factors contributing to RAI refractoriness are yet to be fully elucidated. Hence, the clinical and genomic profiling of RAI-refractory PTCs are essential to guide and develop more effective diagnostic/prognostic biomarkers and alternative therapies.

Mutational signatures can reflect a predisposition towards specific mutations due to active endogenous and/or exogenous biological processes operating on the human cancer genome [14,15]. Accumulated mutations may be resultant of multiple mutagenic processes over the lifespan of a tumor [16]. Mathematical approaches allow individual mutational signatures to be deconstructed from somatic mutation catalogues, whilst approximating the number of mutations with the probability that they were caused by each mutational signature [17,18,19,20,21]. Consequently, mutational processes such as aging and the pathological activation of apolipoprotein B mRNA-editing enzymes (APOBEC) have recently been implicated in the heterogeneous nature of PTC [16,22,23,24]. Importantly, mutational signatures can serve as biomarkers of drug response and disease prognosis, directly impacting patient management and treatment options [25,26,27]. Thus, it is necessary to assess the clinical potential of mutational signatures in RAI-refractory PTC.

This study aims to identify the characteristic clinicopathological and genomic features in RAI-refractory PTCs that can help predict RAI refractory disease early (following primary surgical excision). Whole-exome sequencing (WES) was performed on 158 (66 RAI refractory and 92 avid) adult (>18 years) PTCs and patient-matched germline. RAI refractoriness was defined based on the recently published joint consensus from the American Thyroid Association, the European Association of Nuclear Medicine, the European Thyroid Association, the Society of Nuclear Medicine and Molecular Imaging on Current Diagnostic and Theranostic Approaches, and current literature [6,28,29,30]. All PTC samples were sequenced prior to the classification of RAI refractory disease.

Overall, this study highlights a distinct aggressive clinical nature and select genomic features characteristic of RAI-refractory PTC, whilst proposing the APOBEC SBS13 mutational signature as a novel independent predictor of RAI refractoriness in PTC. The detection of RAI refractory disease following standard surgical removal of PTCs and more suitable therapeutic interventions may potentially improve patient outcomes.

## 2. Materials and Methods

### 2.1. Clinical Cohort

One-hundred and fifty-eight (66 RAI refractory and 92 avid) PTC samples, normal whole blood and clinicopathological data were collected from adult (>18 years) patients, from King Faisal Specialist Hospital and Research Center and Prince Sultan Military Medical City. The tumors were not subjected to any therapeutic regimen prior to surgical excision. Upon optimized surgery, all patients received ≥1 dose of adjuvant RAI therapy. Patients were followed up at 3–6-month intervals for neck ultrasound, thyroid function tests, serum thyroglobulin levels and thyroglobulin antibody analyses. Progression-free survival was defined as the time, in months, from the date of the initial surgery to the detection of any tumor recurrence (local, regional, or distant). In case of no recurrence, the date of last follow-up was considered the study endpoint. Clinical and treatment data were collected from medical records and are summarized in Appendix A.

This study was conducted in accordance with the Declaration of Helsinki and approved by the Institutional Review Board of the King Faisal Specialist Hospital & Research Center, under the Project Research Advisory Committee (RAC) #2110 031 & #2211 168 on PTC archival clinical samples. Written consent was obtained from all patients included in the study.

### 2.2. Imaging Analysis

Whole body ^131^I scans were performed via SPECT (single-photon emission computerized tomography) post-surgery to assess uptake status, and post-^131^I therapy to assess response to therapy. Local recurrences and distant metastases were confirmed by ^131^I scan, CT (computed tomography), MRI (magnetic resonance imaging), ^18^F-FDG PET/CT (fluorodeoxyglucose-positron emission tomography/computed tomography), or bone scan, where relevant.

### 2.3. Definition of Radioactive Iodine Refractoriness

Based on the recently published joint consensus from the American Thyroid Association, the European Association of Nuclear Medicine, the European Thyroid Association, the Society of Nuclear Medicine and Molecular Imaging on Current Diagnostic and Theranostic Approaches, and current literature [6,28,29,30], PTCs were classified as RAI refractory if any of the following were fulfilled:^131^I uptake absent on diagnostic ^131^I scan of locoregional recurrence or distant metastasis.^131^I uptake absent on a ^131^I scan performed several days after ^131^I treatment.^131^I uptake present in some, but not all tumor foci.Disease progression despite a cumulative ^131^I activity of ≥600 mCi.Metastatic disease progression despite ^131^I uptake.Rising serum thyroglobulin levels ≥6 months after ^131^I treatment.Structural disease progression after ^131^I treatment.

### 2.4. Sample Processing

DNA was isolated from formalin-fixed, paraffin-embedded (FFPE) tumor and blood samples using a Gentra DNA isolation kit (Gentra, Minneapolis, MN, USA), following the manufacturer’s recommendations as described previously [31]. DNA was quantified by Qubit (Invitrogen, Waltham, MA, USA).

### 2.5. Whole-Exome Sequencing

Preceding the classification of RAI refractoriness, whole-exome sequencing (WES) was performed using SureSelectXT Target Enrichment (Agilent, Santa Clara, CA, USA) on Illumina NovaSeq 6000, at median depth 198× (range 84–543) for tumors, and 204× (range 94–665) for germline. The sequencing reads were aligned to the human reference genome hg19 using Burrows–Wheeler Aligner (BWA) v0.7.15 algorithm [32], followed by local realignment and PCR duplicate marking via Picard tools (v1.119). Base-quality recalibration was performed with GATK v3.8.0 [33]. All quality metrics were obtained using GATK and FastQC.

### 2.6. Mutation Calling

Somatic single nucleotide variants (SNVs) were identified via MuTect v1.1.7 [34], while somatic small insertions and deletions (indels) were identified via VarScan v2.3.9. Variants were annotated using ANNOVAR [35]. SNVs were filtered using the ”KEEP” parameter, and indels were filtered with a somatic *p*-value filter of ≤0.001. Germline-supporting reads had to be ≥8 with a VAF <5%. Common SNPs with a minor allele frequency of >0.01 in dbSNP, the National Heart, Lung, and Blood Institute exome sequencing project, 1000 Genomes, Exome Aggregation Consortium (ExAC) or in-house exome sequencing data of ~800 normal samples, were excluded. Variants present at segmental duplication regions that had a variant allele frequency (VAF) <5% or a regional sequencing depth of the variant <8 reads were also removed.

### 2.7. Gene Copy Number Profiling

FACETS v0.5.13 [36] was used to determine copy number variants (CNVs), purity and regions of loss of heterozygosity (LOH), while the outputs generated by MuTect and VarScan were utilized to calculate mean allelic frequency (MAF). The CNVs were further defined as gains, losses, amplifications and deletions in relation to average ploidy [37]. CNVs with <10 total markers, and <5 heterozygous markers were excluded from the analysis.

### 2.8. Telomerase Reverse Transcriptase Promoter Mutations

Although the *telomerase reverse transcriptase* promoter (*TERT*p) region is not covered in WES, *TERT*p hotspot mutations were explored using polymerase chain reaction (PCR) and Sanger sequencing, as previously described [38]. All tumors were included, except three cases (Appendix A) for which analysis could not be performed due to insufficient DNA availability. The sequencing traces were analyzed using Mutation Surveyor v5.0.1 (Soft Genetics, LLC, State College, PA, USA). NCBI GenBank reference sequences were used for the mutation analysis.

### 2.9. Driver Events

All nonsilent mutations and CNVs encompassing genes were assessed for pathogenicity using the Cancer Genomics Interpreter via the Thyroid papillary (“THP”) cancer type [39]. Mutations identified as “TIER 1” due to well-documented evidence of promoting oncogenic transformation in cancer or “TIER 2” with strong but still emerging evidence of having roles in cancer, and CNVs categorized as known driver amplifications or deletions in cancer-related genes, were all classified as driver events. All putative driver mutations were manually validated using Integrated Genomics Viewer (IGV) v2.4.10. Variants with >2 mapping quality zero reads were considered false positives and removed from the analysis. Hotspot *TERT*p mutations were also considered driver mutations due to their known role in PTC tumor progression [40,41,42].

### 2.10. Mutational Signature Analysis

Mutational signatures were predicted via the deconstructSigs package in R [21], using the published Catalogue of Somatic Mutations in Cancer (COSMIC) 30 single base substitution (SBS) mutational signatures [43] as a reference. The tumors were initially assessed for all 30 SBS mutational signatures and for associations with RAI avidity. Thereafter, only 20 COSMIC mutational signatures were selected for subsequent analyses due to known biological relevance [16,22,23,24] or high levels and/or prevalence in the analyzed PTC cohort (*n* = 113) (Appendix A).

### 2.11. Statistical Analysis

All statistical analyses were executed on IBM SPSS Statistics (v.21). The Mann–Whitney U Test, the Chi-Squared Test or Fisher’s Exact Test were utilized to compare the continuous and categorical variables, where relevant. Spearman’s rank correlation was used to determine associations. Binary logistic regression modelling was used for the RAI refractory predictive analysis via the “Enter” method in SPSS, where the nonsilent tumor mutation burden (TMB; high/low), mutational signatures (present/absent), *BRAF* (present/absent) and *TERT*p mutation (present/absent) were taken as categorical variables. High TMB was defined as a TMB > the cohort median of 8, whereas a TMB ≤ cohort median was considered low. For all the statistical tests performed, the analyses were two-tailed, with *p* < 0.05 considered statistically significant. For the multivariable analyses (unless specified otherwise), the *p*-values were adjusted for confounding variables, including age, tumor size and tumor purity, as continuous variables. To decrease the false discovery rate (FDR), the *p*-values were corrected using the Benjamini–Hochberg method adjusted at FDR <10%, where relevant.

## 3. Results

### 3.1. Clinicopathological Associations

Potential clinicopathological features associated with RAI refractoriness were investigated using multivariable analysis in the 66 RAI-refractory versus 92 avid PTCs. Although RAI-refractory patients appeared to be slightly older in age (median 44, range 20–83) than avid patients (median 39, range 20–75), this was not statistically significant (*p* = 0.327; Mann–Whitney U Test) (Table 1). In contrast, RAI-refractory tumors were significantly associated with positive surgical margins and lymph node metastases present at the time of diagnosis (*p* = 0.016 and 0.012, respectively; Chi-Squared Test). No other associations were observed.

### 3.2. Genomic Landscape

A total of 1654 somatic nonsilent mutations (median 8, range 0–161) and 162 copy number variants (CNVs) (median 0, range 0–27) were identified across the PTC cohort (Appendix A). In RAI-refractory PTCs, a median of 11 mutations (range 1–161) and a median of 0 CNVs (range 0–10) were identified, whereas RAI-avid PTCs had a median of 7 mutations (range 1–19) and 0 CNVs (range 0–27) (Appendix A). Although the CNV burden did not differ (*p* = 0.797; Mann–Whitney U test), a significantly higher tumor mutational burden (TMB) was observed in RAI-refractory compared to avid tumors (*p* < 0.0001; Mann–Whitney U Test). Despite mutations significantly correlating with age (rho = 0.494, *p* < 0.0001; Spearman’s rank correlation), the significantly higher TMB in RAI-refractory tumors was retained upon multivariable analysis (*p* = 0.011) (Figure 1A).

While CNVs were sparse in both RAI-refractory and avid PTCs, amplifications on chromosome 1q were found exclusively in RAI-refractory tumors (4.5%, 3/66; *p* = 0.076; Fisher’s Exact Test), and deletions on chromosome 2q32.1 were only found in RAI-avid tumors (4.3%, 4/92; *p* = 0.146; Fisher’s Exact Test) (Appendix A).

Putative driver events (mutations and CNVs) were determined in the PTC cohort. A total of 248 driver events were identified in 118 patients (median 1, range 0–60) (Appendix A). On average, significantly more driver events were found in RAI-refractory (1.56 ± 1.67) compared to avid tumors (0.93 ± 0.74) (*p* = 0.010; Mann–Whitney U Test). Driver mutations in *BRAF* (48.5%, 32/66 RAI refractory versus 53.3%, 49/92 avid; *p* = 0.553; Chi-Squared Test) and *RAS* (7.6%, 5/66 RAI refractory versus 10.9%, 10/92 avid; *p* = 0.487; Chi-Squared Test) were amongst the most recurrent and mutually exclusive in both RAI-refractory and avid tumors. Whereas *TERT*p mutations were significantly enriched in RAI-refractory tumors (31.8%, 21/66 versus 2.2%, 2/92 avid; *p* < 0.0001; Fisher’s Exact Test). To test previously reported *BRAF*^V600E^ and *TERT*p mutation (in combination and alone) [44] as predictors of RAI refractoriness, the tumors were divided according to genotype (Appendix A). By *BRAF*^V600E^ alone, RAI refractoriness was only detected in 29.2% (odds ratio [OR] 0.7, 95% confidence intervals [CI] 0.3–1.4) of tumors, by *TERT*p mutation alone in 100% (OR 31.7, 95% CI 1.8–567.5), and by concurrent *BRAF*^V600E^ and *TERT*p mutations in 85.7% (OR 11.8, 95% CI 2.5–56.1), versus 37.3% (25/67) of tumors negative for both markers (*p* = 0.326, 0.019 and 0.002, respectively). 

In contrast, *AKT3* and *MDM4* amplifications were the only driver CNV events found in RAI-refractory PTCs (both in 4.5%, 3/66 of cases; *p* = 0.076; Fisher’s Exact Test), and only a single RAI-avid tumor (PTC-501) exhibited multiple driver CNV events (*p* = 1.000; Fisher’s Exact Test).

### 3.3. Frequency of Mutational Signatures

The mutational processes active in RAI-refractory and avid PTCs were investigated using the 30 published single-base substitution (SBS) mutational signatures from the Catalogue of Somatic Mutations in Cancer (COSMIC) database [43]. Due to the relatively low mutational burden of PTC compared to other cancer types [24], mutational signature analysis could only be performed on 81.8% (54/66) of RAI-refractory and 64.1% (59/92) of avid tumors with ≥10 single nucleotide variants (SNVs) (Appendix A). Overall, the median number of SNVs in the 113 tumors was 20 (range 10–345), with 24 (range 10–345) in RAI-refractory and 17 (range 10–36) in RAI-avid tumors (Figure 2A).

The number of SNVs contributing to each mutational signature was investigated in the RAI-refractory and avid PTCs. No mutational signatures were found to be enriched in RAI-avid tumors. However, SNVs in RAI-refractory tumors were significantly enriched for age-related mutations (SBS1) and APOBEC activity, via both SBS2 and SBS13 (Benjamini–Hochberg-adjusted *p* < 0.001, 0.030 and <0.001, respectively; Mann–Whitney U Test) (Figure 1B).

In contrast with a previous study [24], APOBEC mutational signatures were not associated with *BRAF*^V600E^ mutation. A total of 60.9% (14/23) of tumors were found harboring concurrent APOBEC mutational signature (SBS2 and/or SBS13) and *BRAF*^V600E^ mutations, compared to 52.5% (47/90) of the APOBEC-negative tumors found to be harboring a *BRAF*^V600E^ mutation (*p* = 0.473; Chi-Squared Test). Similarly, APOBEC mutational signatures were not associated with *RAS* mutation (APOBEC and *RAS*-positive tumors: 8.7%, 2/23 versus APOBEC-negative but *RAS*-positive tumors: 20%, 18/90; *p* = 0.364; Fisher’s Exact Test). In contrast, APOBEC mutational signatures were significantly associated with *TERT*p mutation, with concurrent APOBEC and *TERT*p mutations being harbored in 34.8% (8/23), compared to 14.4% (13/90) of APOBEC-negative tumors harboring a *TERT*p mutation (*p* = 0.025; Chi-Squared Test). To determine whether APOBEC-related mutagenesis was the biological process responsible for inducing driver mutations in APOBEC-positive tumors, a probabilistic approach [25] was used. From the 45 driver mutations found in the APOBEC-positive tumors, only seven (15.6%) in three samples comprised of C > T or C > G substitutions in a TCN trinucleotide context. Upon analysis, only four of the seven driver mutations showed a high probability of being induced by APOBEC activity. Notably, the driver mutations *MED12*^S745L^ c.C2234T (97.1% probability; in case THY-380), *PPP2R1A*^S296F^ c.C230T (90.6% probability; in case THY-380) and *STAG2*^Q773X^ c.C2317T (88.5% probability; in case PTC-212) were highly likely to have been induced by APOBEC SBS2 activity. Driver mutation *FXR1*^S400X^ c.C1199G (in case THY-380) was highly likely, with 100% probability, to have been induced by APOBEC SBS13. Therefore, only two cases were likely to have incurred driver mutations as a direct result of aberrant APOBEC activity. Most driver events, including *TERT*p (C228T and C250T) and *BRAF*^V600E^ mutations, were most likely consequent of other biological processes or selection, not APOBEC activity.

### 3.4. Patterns of Mutational Signatures

Using unsupervised clustering, the overall patterns of mutational signatures between RAI-refractory and avid PTCs were investigated. Ten clusters were identified: four main clusters (C1–C4) and six relatively smaller clusters (C5–C10) (Figure 2B). Cluster C1 predominantly exhibited the clock-like SBS5 mutational signature of unknown etiology. Cluster C2 was characterized by the dominant homologous recombination-based DNA repair (HR; SBS3) mutational signature. Cluster 3 mainly contained age-related mutations (SBS1). Cluster 4 presented with relatively high contributions of SBS16 mutations (of unknown etiology). Cluster 5 involved a high proportion of mutations related to aflatoxin exposure (SBS24). Cluster C6 contained tumors with a mix of defective DNA mismatch-repair (MMR) signatures, including SBS6, SBS15, SBS20, SBS21 and/or SBS26. Cluster C7 predominantly presented the defective DNA base excision repair (BER; SBS30) mutational signature. Two very small clusters showed high contributions of SBS8 (C8) and SBS19 (C9) mutations, both of unknown etiology. Cluster 10 was distinctly characterized by the APOBEC mutational signatures SBS2 and SBS13. Furthermore, while clusters C1–C9 were found in both RAI-refractory and avid tumors, C10 was the principal cluster found in RAI-refractory tumors, which was also coupled with concurrent *TERT*p mutations (in 55.6%) and higher TMBs (in 77.8%).

### 3.5. Prediction of RAI Refractoriness 

Multivariable logistic regression was used to assess if any of the mutational signatures, TMB, *BRAF*^V600E^ or *TERT*p mutations could be used to predict RAI refractoriness independently in PTC. Only the APOBEC SBS13 mutational signature (OR 30.4, 95% CI 1.4–647.2) and *TERT*p (OR 41.3, 95% CI 4.4–391.6) mutations significantly predicted RAI refractoriness in PTC (*p* = 0.029 and 0.001, respectively) (Figure 3). In contrast, *BRAF*^V600E^ (OR 0.3, 95% CI 0.1–1.0) and the mutational signature SBS23 of unknown etiology (OR 0.2, 95% CI 0.03–0.9) mutations significantly predicted a lower risk for developing RAI refractory disease in our PTC cohort (*p* = 0.043 and 0.034, respectively). 

To further investigate the clinical utility of the APOBEC SBS13 mutational signature and *TERT*p mutation as independent biomarkers to predict RAI refractoriness in PTC, the cohort was divided according to genotype (Table 2). RAI refractoriness was predicted in 87.5% (OR 14.5, 95% CI 1.7–124.0) of tumors with SBS13 alone, 85.7% (OR 12.4, 95% CI 2.6–59.6) of tumors with *TERT*p mutation alone, and 100% (OR 30.8, 95% CI 1.7–559.4) of tumors harboring concurrent SBS13 and *TERT*p mutations, versus 32.5% (27/83) of tumors negative for both markers (*p* = 0.015, 0.002 and 0.021, respectively).

## 4. Discussion

In a relatively large cohort, the clinical and genomic landscape of RAI-refractory and avid PTCs were systematically profiled, revealing important differences in clinicopathological presentation, TMB, *TERT*p mutation and mutational signatures. 

Whereby, the clinicopathological presentation of Saudi RAI-refractory PTC patients was noticeably more aggressive than avid patients, with multivariable analysis showing significant involvement of surgical margins and presence of lymph node metastases at the time of diagnosis in RAI-refractory PTCs. This was in accordance with RAI-refractory PTC patients of African ancestry [28]. In contrast, Shobab et al. [45] did not find any differences between RAI-refractory and avid patients, possibly be due to their smaller sample size and/or differences in population ethnicity, with >50% of their cohort being of “white” ethnicity.

Despite PTC having a widely accepted lower TMB than most cancers, the average TMB of the PTC cohort was still lower (0.25 nonsynonymous mutations/Mb) compared to TCGA-Thyroid (The Cancer Genome Atlas; 0.41 nonsynonymous mutations/Mb) and previous reporting [46,47]. However, as mutations correlate with age in PTC [24], this can partially be explained by a younger median age at diagnosis of PTC in the Saudi population [3]. Nevertheless, a significantly higher TMB was observed in RAI-refractory compared to avid tumors, though TMB was not found to be an independent predictor of RAI refractoriness.

Notably, the APOBEC SBS13 mutational signature and previously identified *TERT*p mutations [48,49] were found to be highly predictive as independent biomarkers of RAI refractoriness in a distinct aggressive subgroup of PTC in our cohort, whereas *BRAF*^V600E^ and SBS23 mutations were more predictive of RAI avidity rather than refractoriness. Even when only considering *BRAF*^V600E^ and *TERT*p mutations (alone or in combination), only *TERT*p mutations alone were predictive of RAI refractoriness in PTC. In contrast to previous reporting [44], *BRAF*^V600E^ instead suppressed the predictive power when co-existing with *TERT*p mutation. However, the status of SBS13 was not considered by the authors. Despite reports of mutated *BRAF*^V600E^ in RAI-refractory patients, the effects of this mutation are inconsistent. Some studies report repression of the plasma glycoprotein involved with iodide uptake, sodium iodine symporter (NIS), due to *BRAF* mutations in PTCs [50,51], suggesting *BRAF* mutation is associated with RAI-refractory PTC. Other studies revealed no significant difference in *BRAF* mutations between RAI-refractory and RAI-avid PTC patients [52,53]. It is known that the genetic alterations involved in PTC tumorigenesis vary across different ethnicities, indicating that genetic and environmental backgrounds can influence the prevalence and effect of genetic alterations in a population. In our study, *BRAF* mutation was more predictive of RAI avidity than refractoriness, which might partly be explained by differences in patient ethnicity and environmental backgrounds, but this needs to be validated in a larger study. 

Furthermore, in contrast to previous reporting [24], APOBEC mutational signatures were not associated with *BRAF*^V600E^ mutation in our PTC cohort, but were significantly associated with *TERT*p mutation. APOBEC mutagenesis, specifically through kataegis (when mutations cluster at rearrangement breakpoints) during telomere crisis can promote tumor growth, but not indefinitely [54]. However, when combined with *TERT*p, TERT re-activating mutations can sustain cellular lifespan by healing short telomeres [55], to help restore their capacity for DNA proliferation [56]. Whether APOBEC mutations were occurring via kataegis clustering or throughout the genome in cases with concurrent *TERT*p could not be determined due to the limited scope of this study. Future whole-genome mutational signature analysis and functional studies would be required.

Since only SBS13 and *TERT*p were found to be positive predictors of RAI refractoriness, we focused on testing their ability alone or in combination to predict RAI refractoriness. Both SBS13 and *TERT*p mutations alone highly predicted RAI refractoriness in PTC. However, when these two markers co-existed, the likelihood of predicting RAI-refractory PTC was significantly increased.

The SBS13 APOBEC mutational signature produces distinctive C > G and C > A mutations (in a TCN trinucleotide context) generated by the uracil excision of error-prone polymerases or abasic sites by uracilDNA glycosylase, in contrast to the characteristic C > T mutations of SBS2, which are triggered by uracil replication following APOBEC cytidine deamination [43,57]. 

As previously reported [22,24,57], SBS2 and SBS13 in our PTC cohort were also observed in TMB-high thyroid cancers. RAI-refractory tumors formed a unique cluster enriched with APOBEC mutational signatures, high TMB and *TERT*p mutations.

As standard RAI therapy may not be appropriate for all PTC patients, the development of further treatment options is imperative. As such, ATR (ataxia telangiectasia and Rad3-related kinase) and immune-checkpoint inhibitors could potentially be alternatives for the majority of our RAI-refractory tumors, due to the collective high TMB and/or involvement of aberrant APOBEC activity, via synthetic lethality [25,26,58]. It would be relevant for such patients to be included in clinical trials investigating the benefit of these therapies in RAI-refractory PTC.

Due to the uniqueness of our study and the controversy surrounding the definitions of RAI refractoriness in PTC, the available scientific literature for direct comparison was limited. Still, despite the need of further validation, our findings may help identify patients more likely to be refractory to standard RAI therapy based on their mutational signatures obtained from WES data of routinely excised PTC tumors.

## 5. Conclusions

This study highlights distinct clinical and genomic features in RAI-refractory PTC, proposing the APOBEC SBS13 mutational signature as a novel independent predictor of RAI refractoriness in a more aggressive subgroup of PTC. Where the detection of RAI refractory disease following the standard surgical removal of PTCs, and more suitable therapeutic interventions may potentially improve patient outcomes.

## Figures and Tables

**Figure 1 cancers-14-01584-f001:**
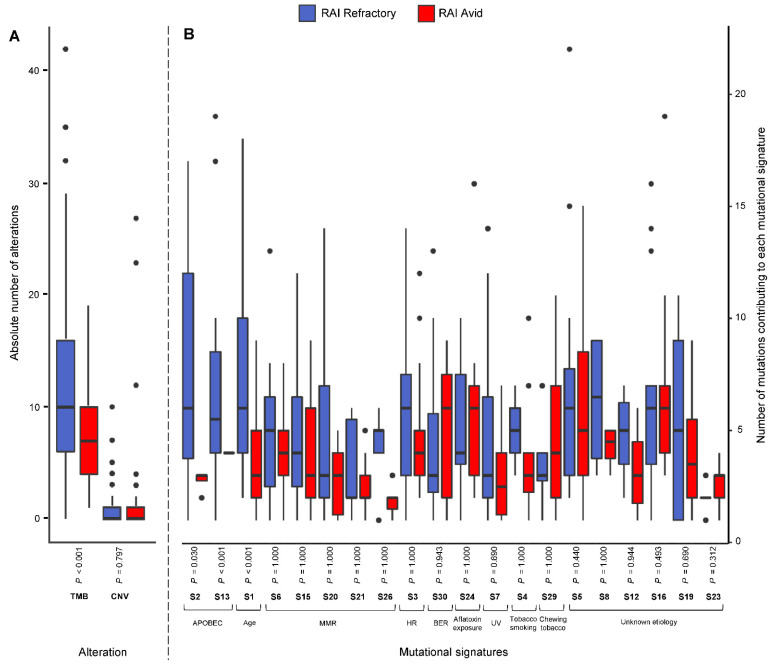
Box plots showing (**A**) the absolute number of alterations, including mutations (tumor mutation burden; TMB) and copy number variations (CNV) in RAI-refractory and avid PTC tumors, and (**B**) the number of mutations contributing to selected single-base substitution (S; SBS) mutational signatures, with Benjamini–Hochberg-adjusted *p*-values (Mann–Whitney Test; alpha level 0.05). For each SBS mutational signature, only tumors with ≥1 mutation contributing to the SBS mutational signature were included for box plot representation. S1 represents age-related mutations; S2 and S13 represent APOBEC-related mutagenesis; S6, S15, S20, S21 and S26 represent defective DNA mismatch-repair (MMR) mutations; S3 represents defective homologous recombination-related DNA repair (HR); S30 represents defective base-nucleotide excision repair (BER); S4 represents tobacco smoking; S7 represents ultraviolet (UV) exposure; S24 represents aflatoxin exposure; S29 represents chewing tobacco; and S5, S8, S12, S16, S19 and S23 are of unknown etiology.

**Figure 2 cancers-14-01584-f002:**
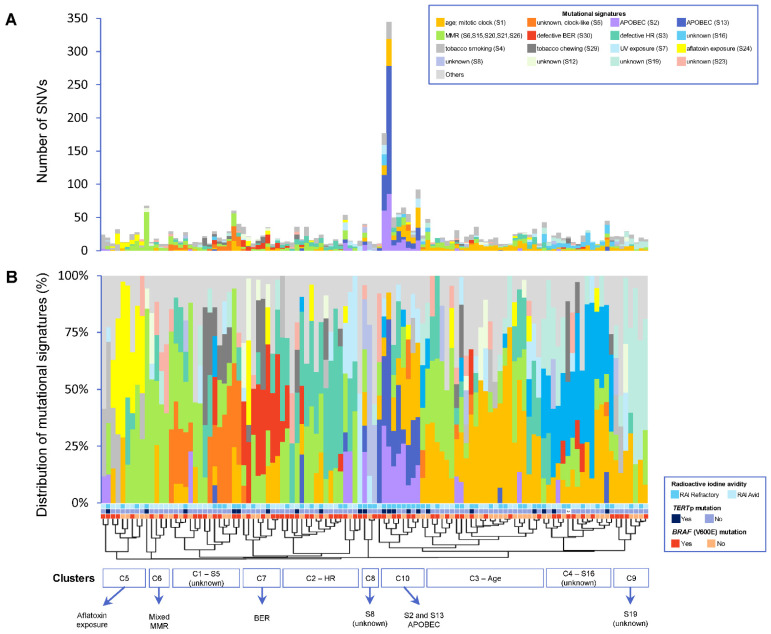
(**A**) Overview of absolute number of mutations contributing to each mutational signature per sample. (**B**) Distribution of each mutational signature per sample. Patients are ordered by hierarchical clustering. An alpha level of 0.05 was used for statistical significance. S1 represents age-related mutations; S2 and S13 represent APOBEC-related mutagenesis; S6, S15, S20, S21 and S26 represent defective DNA mismatch-repair (MMR) mutations; S3 represents defective homologous recombination-related DNA repair (HR); S30 represents defective base-nucleotide excision repair (BER); S4 represents tobacco smoking; S7 represents ultraviolet (UV) exposure; S24 represents aflatoxin exposure; S29 represents chewing tobacco; and S5, S8, S12, S16, S19 and S23 are of unknown etiology.

**Figure 3 cancers-14-01584-f003:**
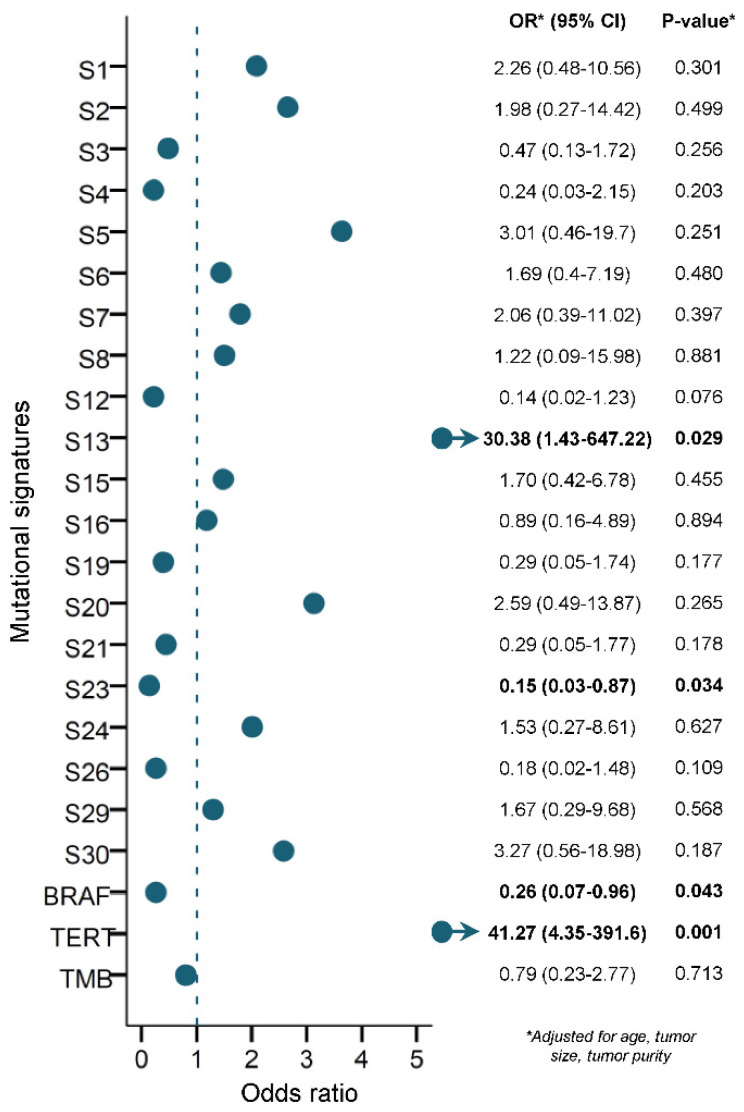
Multivariable logistic regression model (when adjusting for age, tumor size and tumor purity as continuous variables), using mutational signatures, *BRAF*^V600E^ mutation, *TERT* promoter mutation and TMB to predict RAI refractoriness in PTC. An alpha level of 0.05 was used for statistical significance. S1 represents age-related mutations; S2 and S13 represent APOBEC-related mutagenesis; S6, S15, S20, S21 and S26 represent defective DNA mismatch-repair (MMR) mutations; S3 represents defective homologous recombination-related DNA repair (HR); S30 represents defective base-nucleotide excision repair (BER); S4 represents tobacco smoking; S7 represents ultraviolet (UV) exposure; S24 represents aflatoxin exposure; S29 represents chewing tobacco; and S5, S8, S12, S16, S19 and S23 are of unknown etiology.

**Table 1 cancers-14-01584-t001:** Clinicopathological characteristics of radioactive iodine (RAI) refractory (*n* = 66) and avid (*n* = 92) PTC cases.

Clinicopathological Variables	RAI Refractory *n* (%)	RAI Avid *n* (%)	Odds Ratio (95% CI ^1^)	Multivariable *p*-Value
**Age**				0.327
Median	44	39	1.0 (0.9–1.0)	
Range	20–83	20–75		
**Sex**				0.483
Male	18 (27.3)	21 (22.8)	0.7 (0.2–2.1)	
Female	48 (72.7)	71 (77.2)		
**Histopathological subtype**				
Classical variant	33 (55.9)	45 (57.7)	1	
Follicular variant	12 (20.3)	24 (30.8)	0.9 (0.2–4.6)	0.918
Tall cell variant	14 (23.7)	9 (11.5)	0.9 (0.2–3.9)	0.932
**Extrathyroidal extension**				0.155
Absent	26 (39.4)	63 (68.5)	1	
Present	40 (60.6)	29 (31.5)	4.5 (0.6–35.7)	
**Lymphovascular invasion**				0.239
Absent	25 (50.0)	47 (55.3)	1	
Present	25 (50.0)	38 (44.7)	0.5 (0.2–1.6)	
**Surgical margin involvement**				0.016
Absent	20 (34.5)	66 (73.3)	1	
Present	38 (65.5)	24 (26.7)	4.8 (1.3–17.0)	
**Tumour focality**				0.577
Unifocal	39 (59.1)	41 (44.6)		
Multifocal	27 (40.9)	51 (55.4)	1.4 (0.5–4.1)	
**Stage**				
I	34 (51.5)	65 (70.7)	1	
II	7 (10.6)	9 (9.8)	0.3 (0.01–8.3)	0.484
III	10 (15.2)	13 (14.1)	2.2 (0.3–15.0)	0.413
IV	15 (22.7)	5 (5.4)	3.3 (0.1–72.2)	0.453
**pT**				
T1	8 (12.1)	30 (32.6)	1	
T2	12 (18.2)	23 (25.0)	1.2 (0.2–6.3)	0.864
T3	35 (53.0)	36 (39.1)	0.7 (0.1–6.2)	0.771
T4	11 (16.7)	3 (3.3)	0.5 (0.01–24.3)	0.745
**pN**				0.012
N0	24 (37.5)	58 (69.9)	1	
N1	40 (62.5)	25 (30.1)	4.4 (1.4–14.3)	
**pM**				
M0	54 (81.8)	91 (98.9)		
M1	12 (18.2)	1 (1.1)		
**Types of surgery**				0.059
Hemi-thyroidectomy	6 (9.1)	7 (7.6)	1	
Total thyroidectomy	60 (90.9)	85 (92.4)	0.1 (0.01–1.1)	

^1^ CI: confidence interval.

**Table 2 cancers-14-01584-t002:** Relationship between the APOBEC SBS13 mutational signature alone, *TERT* promoter (*TERT*p) mutation alone, their co-occurrence, and radioactive iodine (RAI)-refractoriness in PTC.

Genotype	Total Cases *n*	RAI Refractory *n* (%)	RAI Avid *n* (%)	*p*-Value (Alpha = 0.05)	Odds Ratio (95% Confidence Intervals)
SBS13 (−) and *TERT*p (−)	83	27 (32.5)	56 (67.5)		
SBS13 (+) and *TERT*p (−)	8	7 (87.5)	1 (12.5)	0.015	14.5 (1.7–124.0)
SBS13 (−) and *TERT*p (+)	14	12 (85.7)	2 (14.3)	0.002	12.4 (2.6–59.6)
SBS13 (+) and *TERT*p (+)	7	7 (100.0)	0 (0.0)	0.021	30.8 (1.7–559.4)

SBS13: single base substitution (SBS) mutational signature 13 mutations; *TERT*p: *TERT* promoter mutations; (−): negative for/absent; (+): positive for/present.

## Data Availability

The whole-exome sequencing data of this study are part of the International Cancer Genome Consortium (ICGC), which have been deposited to the European Genome-phenome Archive (EGA) under the accession number: EGAS00001001788.

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
