# Peer review of "APOBEC SBS13 Mutational Signature—A Novel Predictor of Radioactive Iodine Refractory Papillary Thyroid Carcinoma"

_cancers, 2022, doi:10.3390/cancers14061584_

Round 1

Reviewer 1 Report

The authors present an interesting and novel study on APOBEC SBS13 Mutational Signature as a new predictive marker in refractory, RAI non-sensitive thyroid papillary carcinomas. The study is well designed. the results are illustrated with several tables and graphs. The discussion is well written. The reference list is up-to-date.

Minor comments:

Are there any influence of genetic background, would that marker be useful also in general population? Just discussion on this is sufficient.

BRAF was very profoundly studied in past - can you comment more in details on your conflicting data, compare with Korean studies.

Reviewer 2 Report

This is a nice paper aiming to find clinicopathological and genomic predictors for RAI response or refractory tumors. Although the overall prognosis of papillary thyroid carcinoma is excellent, RAI refractory patients pose a treatment challenge and harbor worse prognosis that RAI avid PTC patients; Hence, the objective of this study to find predictors for RAI avidity can be the first step towards early recognition of this patients' population.

The authors clearly stated the methods used for the study. The results section was very well written, with an easy and comprehensive flow.

One suggestion that might strengthen this manuscript is to use an independent cohort of patients - that wasn't used in the current analysis-  to validate the results of the integrative data and proposed predictors that were presented in the results section.

Reviewer 3 Report

The manuscript 1608106 to Cancers examines the genetic influence on success/failure of thyroid cancer surgical intervention followed by radioactive I-131 treatment. The authors have found several genes predisposing to refractory character of the treatment. They build a predictive model based on these genetic data based on a medium size group of 154 patients.

The groups of successful and refractory patients are unequal (42% refractory 64/(64+92)), but the difference is acceptable for model building, noting that in treatment only about 5-20% cases are refractory.

The authors identified APOBEC SBS13 mutational signature as a predictor. Previously TERTp mutation has been found as predictive of refractory character.

Major

1. My major concern is in statistical analysis principally in the preparation and analysis of binary logistic regression (BLR). I use SPSS26 and extensively used binary logistic regression in this package. The typical use is for constructing predictive model based on a set of potentially correlated variables. Unfortunately BLR is not used anywhere in the text except Methods. Multivariate is applied in analysis. These two should not be confused. My specific questions and suggestions are:

• There is no mention of type of regression – is it forward or backward and if both end up in the same set of variables used in the model. SPSS gives several options.

• BLR has given two statistically significant (p<0.05) biomarkers S13 and TERT with odds ratio significantly >1 and one protective gene S23 with odds ratio significantly < 1. This is a perfect set of three to construct the model, however there is no model, and no information on correlation of these variables. Why there is no combined model presented? SPSS suggests the use of variables of 0.1>p>0.05, which sometimes strengthens the model. There is one like that in Table 1 and

• There is no analysis of ROC/AUC curves nor values neither of the selected individual predictors nor of the model. Please include them, also giving confidence intervals of AUC and curve with error strips. Such analysis can be done in R and using for example pROC package for R. Please report probabilities curve, not 0/1 ROC.

• For a BLR it is normally beneficiary if variables are continuous, or natural numbers (Scale type in SPSS). Figure 1 uses absolute number of alterations; this may improve the quality of BLR analysis. Try it and report in the manuscript.

• I appreciate sharing of original data in supplementary Excel file. Please include XML model file from SPSS as supplementary file, which table of Excel file it uses and correspondence between SPSS columns and Excel ones.

• Build a model taking random 70% of patients and validate on remaining 30%. Analyze the repeatability.

2. Table 2 and its explanation is difficult to understand. I suggest to apply binary logistic regression there, build a model, put ROC curves, AUC +/- CI values and that discussion will be much more powerful. Authors put n and then use a fraction (n/m). What does None mean there, Other mutations? Cancer patients with no mutations? There are 66 refractory patients and there appears 83 in denominator, what is this value?

3. In Figure 1 S13 value of 4 has no variance, no boxes. This may be the accidental feature responsible for all the success of this variable. Please discuss it.

Minor

• Line 103. Please state which modality was used to acquire I-131 scans. I suppose this was either direct activity measurement or SPECT.

• Figure 1. Explain bars, boxes, points.

• Line 362. TCGA abbreviation is not explained.

Reviewer 4 Report

In this paper the authors present results of sequencing for a large series of RAI refractory thyroid cancer samples and find some expected results with TERT promoter mutations and higher mutational loads and a novel finding of a APOBEC single base substitution COSMIC mutational signature 13 in the RAIR samples which had a higher OR of over 30 in predicting RAIR disease.

This is an interesting and well written study, I would suggest the following minor changes to improve the paper for the reader.

From the text is it correct that the tumour sequencing was performed before a diagnosis of RAIR disease? This is important to highlight and should be included in the introduction and/or abstract if it is correct, was any of the sequencing performed on samples obtained after RAI treatment such as the patients who had recurrence treated surgically.

The median tumor mutational burden (TMB) should be reported in the results given this was used as the cutoff for high/low classification.

Given the extensive clinical info reported, do the authors have any data on overall survival and further treatments such as TKIs given to the patients in the series?

Round 2

Reviewer 2 Report

The authors reply to my comments was satisfactory.